# Evaluation of Eight Genotypes of Corn for the Commercial Cultivation of Huitlacoche in Nopalucan, Puebla, Mexico

**Omar Garcilazo Rahme [1], Isaac Tello Salgado [2], Gerardo Mata [3], Conrado Parraguirre Lezama [4], Maria de los Angeles Valencia de Ita [4] and Omar Romero Arenas [4],***

[1] Posgrado en Manejo Sostenible de Agroecosistemas, Benemérita Universidad Autónoma de Puebla, Puebla 72540, Mexico; ograhme@gmail.com

[2] Centro de Investigaciones Biológicas, Universidad Autónoma de Morelos, Cuernavaca 62209, Mexico; hm_teonanacatl@yahoo.com.mx

[3] Instituto de Ecología, AC, Carretera antigua a Coatepec, No. 351, Xalapa CP. 91070, Mexico; gerardo.mata@inecol.mx

[4] Centro de Agroecología, Instituto de Ciencias, Benemérita Universidad Autónoma de Puebla, Edificio VAL 1, Km 1,7 carretera a San Baltazar Tetela, San Pedro Zacachimalpa, Puebla CP. 72960, Mexico; conrado.parraguirre@correo.buap.mx (C.P.L.); maria.valenciadeita@correo.buap.mx (M.d.l.A.V.d.I.)

* Correspondence: biol.ora@hotmail.com; Tel.: +52-222-229-55-00 (ext. 1317)

**Abstract:** The infection caused by *Ustilago maydis*, commonly called huitlacoche, appears in regions of Mexico that produce corn (*Zea mays*) during seasonal conditions. The infection leads to form galls with high levels of proteins, amino acids, and minerals, providing important benefits to nutrition, and it is also becoming relevant due to its high commercial value, becoming a potential crop for Mexico, Central America, and the United States. The objectives of the present investigation are to evaluate the potential yield per hectare (Kg ha$^{-1}$) based on the incidence percentage (PI), and severity index (ISE) in eight genotypes of corn, as well as performing proximal chemical analysis and mineral element analysis with the galls obtained via inoculation of the MA-Um1 strain of *U. maydis*. The experimental unit is made up of eight subgroups of 250 plants here, considering four hybrid and four Creole genotypes of corn used by producers in open fields in Nopalucan, Puebla, Mexico. The Creole creamy-white corn achieved the highest production of huitlacoche (12,759.21 Kg ha$^{-1}$) here, obtaining an incidence percentage of 73.90%; the highest caloric content (39.90 Kcal per 100 g of fresh mushroom) was recorded in the galls produced from the Asgrow Hawk hybrid, while the highest concentration of zinc (2.33 mg per 100 g) was presented by the AS-722 hybrid.

**Keywords:** edible fungi; *Ustilago maydis*; native corn; hybrid corn; productivity

## 1. Introduction

The dimorphic biotrophic fungi *Ustilago maydis* is the cause of the disease known as corn charcoal [1]. In Mexico, there have been references to its consumption as food since pre-Hispanic times, where the Aztecs called it "cuitlacochtli", a combination of the words "cuitla" (tl) (meaning dirt, garbage, or excrement) and "cochtli" (meaning asleep), and it is considered an exotic dish of Mexican cuisine, with high sources of protein, essential fatty acids, carbohydrates, and phenolic compounds [2,3]. Huitlacoche is the name given to the young, fleshy, edible galls that form when ears of *Zea mays* are infected by *U. maydis* in random conditions, where they depend on the environmental conditions that favor the process of infection and development of the host [4,5]. The most productive period for this

food is concentrated in the months of July and August, obtaining between 300 and 500 tons per year, where production is concentrated in the center of Mexico [6–9].

Studies conducted recently have focused on artificial infection, with galls induced by injection with a concentrated suspension of teliospores or basidiospores in the silk channel [9]. Inoculation is more effective during the emergence of the silk channel (8 to 14 days), presenting more serious infections of *U. maydis* at four and eight days after the appearance of the silk channel for 50% of the inoculated plants [9,10]. However, the suggested inoculation period for selective production is two to four days after the appearance of the silk channel [11].

Corn cultivation in Mexico occupies 57% of the surfaces reserved for cultivating basic grains and oilseeds, where more than 2.5 million farmers are dedicated to cultivating corn [12]. The vast majority of corn genotypes have some degree of resistance to attack by *U. maydis*, with sweet corn genotypes being the most susceptible [13]. The main factors to consider for the commercial production of huitlacoche are the following: (a) the strain used as an inoculant; (b) the period of inoculation after the appearance of silk; and (c) the environmental conditions [9,14]. Likewise, the flavor, aroma, and nutritional value of huitlacoche are all factors that depend on the genotype of corn and the stage of development in which the galls are harvested [15]. Warm and moderately dry areas with average temperatures ranging from 26 to 34 °C, high relative humidity (60–80%), and greater precipitation (43 to 53 mm) increase the frequency of stormy conditions, and this is favorable for the growth of huitlacoche [16,17]. In addition, high amounts of nitrogen and soils fertilized with organic matter predispose plants for better production [4].

Huitlacoche has been characterized as a high-quality nutraceutical mushroom as it contains β-glucans, free sugars, and antimutagenic substances [15], in addition to being used to enrich other foods. Due to its extraordinary flavor and exceptional quality, it is consumed mainly in Mexico, Central America, and the United States, where the cost of huitlacoche for the restaurant market can range from 30 to 40 US dollars for a kilogram of fresh galls [9,11,18].

The evaluation of corn genotypes for the commercial production of huitlacoche allows elucidation of their levels of genetic susceptibility and the ability to select highly productive genotypes for various study areas [19]. The state of Puebla, Mexico, features 515,616 h$^{-1}$ of planted corn and 90% of the corn is planted in areas with seasonal conditions, with an average yield of 1966 Kg ha$^{-1}$. It is important to mention that in the municipality of Nopalucan, Puebla, Mexico, corn production amounted to 17,560 tons from 7689 hectares under seasonal conditions in the 2018 agricultural cycle, amounting to an average yield of 2.28 tons ha$^{-1}$ [20]. Therefore, the objective of this research focuses on evaluating the potential yield per hectare (Kg ha$^{-1}$) based on the incidence percentage (PI) and severity index (ISE) of the MA-Um1 strain of *U. maydis* in eight genotypes of corn (*Zea mays*) in Nopalucan, Puebla, Mexico. In addition, this work features the proximal chemical analysis and mineral element analysis of the galls from the different genotypes of corn. These analyses are carried out in order to obtain knowledge to support the appropriate selection of a suitable genotype for the commercial production of huitlacoche in seasonal crops, taking advantage of the increasing demand and its great potential in the international market.

## 2. Materials and Methods

### 2.1. Study Area

The experiment was carried out in seasonal corn cultivation plots during the 2018 agricultural cycle in Nopalucan, Puebla, Mexico (19°12′59″ N and 97°49′19″ W), at an altitude of 2300 m a.s.l. The region is adjacent to the north with the state of Tlaxcala, the municipalities of San José Chiapa, Rafael Lara Grajales, and San José Chiapa; to the east with the municipalities of San José Chiapa, Rafael Lara Grajales, San José Chiapa, Mazapiltepec de Juárez, and Soltepec; to the south with the municipalities of Soltepec, Acatzingo, Tepeaca, and Acajete; and to the west with the municipality of Acajete and the state of Tlaxcala. In the area, soil types predominate as follows: Regosol (31%),

Durisol (17%), Luvisol (14%), Fluvisol (14%), Phaeozem (11%), Solonchak (5%), and Leptosol (3%). The climatic conditions of the region feature an average temperature of 16 °C, annual rainfall ranging from 500 to 900 mm, and a temperate sub-humid climate with rainfall in summer [21].

### 2.2. Biological Material

The MA-Um1 strain of *U. maydis* is protected in the Ceparium of Edible Mushrooms of the Agroecology Center of the Autonomous University of Puebla, where samples are kept at −70 °C in 50% (*v/v*) glycerol [22].

Yeasts (*Ustilago maydis*) of the diploid strain MA-Um1 were used here, where they were increased in a Holliday medium with 0.05 mg/L of chloramphenicol to avoid contamination. The yeasts were incubated at 26 °C in the dark for a period of 10 days. Once growth was present, the yeasts were transferred to a potato dextrose agar (PDA) medium under the same temperature conditions for 10 days [23].

Two flasks with 75 mL of corn extract each were prepared, sterilized at 15 pounds of pressure for 15 min, inoculated with 0.05-mm slices of the MA-Um1 strain grown in the PDA medium, and incubated by constant shaking at 28 °C for 96 h. In particular, the first flask was useful for counting cells (basidiospores) and confirming the growth and viability of the MA-Um1 strain. For the first case, once the stirring time was completed, 1.50-mL samples of each were taken and placed in Eppendorf tubes, then centrifuged at 14,100 rpm for two minutes, where the supernatants were then discarded and the pellets were homogenized with 1 mL of sterile NaCl at 4%. Then, five 200-μL samples were seeded on corn extract agar plates and incubated at 28 °C until growth was obtained. The second flask was considered to be where the stock solution was obtained. The volume formula used here established the amount of sterilized water that was required for the inoculation field at a concentration of $1 \times 10^6$ basidiospores/mL, which was then stored at 4 °C for 24 h before application. It is worth mentioning that the concentrations used in the field were adjusted with the help of a hematocytometer, and growth and viability were corroborated according to the methodology used by Villanueva et al. [17].

### 2.3. Corn Genotypes

The experimental unit was made up of eight subgroups of 250 plants, considering four hybrid genotypes of corn used by producers in the area: Hawk and Sparrow-hawk marketed by Asgrow, AS-722 of Aspros, and Z-60 by Hartz seed, and four genotypes of Creole corn (white, creamy-white, yellow, and blue), provided in the same way by the producers of the region, obtaining a total of 2000 corn plants for the field experiments. In addition, 50 plants of each genotype were considered as control groups.

### 2.4. Field Experiment

The experiment was established with a completely randomized design, considering eight treatments plus the control group, where each experimental unit was made up of 10 m-long grooves that were located 85 cm apart. The experiment followed the methodology of Madrigal-Rodriguez et al. [24], which relates to the time of the inoculation, where corn cobs should be inoculated in the baby corn stage known as "jilote" (outbreaks silks or early phenological stage R1), which was reached on 16 August 2018. Overall, 2000 plants were inoculated four days after the appearance of the silk canal, applying 1 mL of the inoculum by means of a Vet-Matic® syringe at two equidistant points [9,22]. In the case of the control group, 400 plants were inoculated on the same date, applying 1 mL of a sterile physiological saline solution through a Vet-Matic® syringe at two equidistant points along the baby corn. The sowing density used by the producers in the study area corresponds to 60,000 corn plants per hectare, using the fertilization dose 240-60-60 at the time of sowing (16 June 2018), with 50% nitrogen and 100% phosphorus and potassium, where the other half of nitrogen was applied 45 days after sowing. In order to keep the cultivation plots free of weeds, atrazine (25%) and metolachlor

(25%) were applied in doses of 4 L ha$^{-1}$ and two mechanical weeding operations were carried out (20 and 40 days after sowing). During the development of the experiment (30 days), the maximum (T. max) and minimum (T. min) temperatures were recorded daily, as well as the precipitation of the meteorological station San José Ovando 26.032 of the municipality of Nopaluca, Puebla, Mexico [25].

Huitlacoche was harvested 30 days after inoculation in the different genotypes of corn. This process was carried out with corn that present a black mass, called "gall" [26]. Once collected, a cut was made in the bracts at the insertion point of the peduncle to avoid mechanical injuries that would induce further oxidation and deterioration in the galls obtained 30 days post-inoculation [15]. The control group did not present galls in the eight inoculated genotypes. Once the harvesting of the huitlacoche was complete, data collection was carried out to evaluate the net weight, leaf weight, kernel weight, weight of the shelled mushroom or grams of galls produced by infected cobs (GMI), the presence in infected plants, and also corn cob kernels without galls. For the severity measurement (SEV$_i$), the proportion of the corn covered with the galls of *U. maydis* was considered, as mentioned by Villanueva et al. [17], where five degrees of severity were defined for the development of galls on the corn (Table 1).

**Table 1.** Severity of infection by the MA-Um1 strain according to the percentage of coverage present on the corn cob.

| Category * | Cob Cover by Galls (%) | Judgment of Categorization |
|:---:|:---:|:---:|
| SEV$_1$ | 0 | No presence of galls. |
| SEV$_2$ | >0–25 | Less than or equal to 1/4 length of corn cob with the presence of galls. |
| SEV$_3$ | >25–50 | Greater than 1/4 and less than or equal to 1/2 the length of the corn cob with the presence of galls. |
| SEV$_4$ | >50–75 | Greater than 1/2 and less than or equal to 3/4 the length of the corn cob with the presence of galls. |
| SEV$_5$ | >75–100 | Greater than 3/4 and full coverage of the length of the corn cob with the presence of galls. |

* SEVi = degree of severity present in the inoculated plants.

The incidence percentage (PI) was obtained by dividing the total of infected corn cobs according to the severity category in the experimental unit by the total inoculated corn cobs multiplied by 100 [24] for each genotype evaluated Equation (1).

$$PI = [((\text{number of corn cobs with severity 1}) + (\text{number of corn cobs with severity 2}) + (\text{number of corn cobs with severity 3}) + (\text{number of corn cobs with severity 4}) + (\text{number of corn cobs with severity 5}))/TEI] \times 100 \tag{1}$$

where PI denotes the percentage of incidence and TEI denotes the total inoculated corn cobs.

The severity index (ISE) was obtained based on the work of Villanueva et al. [17] through the following expression Equation (2):

$$ISE = [((\text{number of corn cobs with severity } 1 \times (0)) + (\text{number of corn cobs with severity } 2 \times (0.25)) + (\text{number of corn cobs with severity } 3 \times (0.50)) + (\text{number of corn cobs with severity } 4 \times (0.75)) + (\text{number of corn cobs with severity } 5 \times (1))/TEIf)))))] \times 100 \tag{2}$$

where ISE denotes the severity index (expressed as a percentage) and TEIf denotes the total infected corn cobs.

The potential yields per hectare of huitlacoche were calculated using the methodology of Madrigal-Rodriguez et al. [24] Equation (3):

$$RPH = [(PI \times DS) \times (GMI)]/1000 \tag{3}$$

where RPH denotes the potency yield of huitlacoche per hectare (Kg ha$^{-1}$); PI denotes the percentage of incidence; GMI denotes grams per infected corn cob; and DS denotes the density of plants per hectare used in Nopalucan (60,000).

*2.5. Proximal Chemical Analysis of Huitlacoche of Different Genotypes of Corn*

The proximal chemical analysis was performed following the procedures recommended by the regulations of the Association of Official Analytical Chemists [27,28]. The samples were ground to a particle size of 0.5 to 1 cm. They were oven-dried at 55 °C for 48 h. Crude protein (PC) determination was carried out with dehydrated residues with the micro-Kjeldahl method [29], with a conversion factor of N × 4.28 [27,28]. Lipid determination was carried out by Goldfish AOAC technique No. 95,402 [27,28]. The ashes were determined through incineration at 550 °C [27], while the raw fiber was found by the Van Soest method [30]. The dry matter (DM) was maintained for 36 h at a temperature of 55 °C (DM = [final weight of the sample/initial weight of the sample] × 100) [31]. The contents of carbohydrates present in the samples were calculated using the method of total carbohydrates (TC) by difference [32]. This value was calculated according to the following expression Equation (4):

$$TC = 100 − (\% \text{ water} + \% \text{ protein} + \% \text{ grease} + \% \text{ dietary fiber} + \% \text{ ash} + \% \text{ ethanol}) \tag{4}$$

where TC denotes total carbohydrates.

The energy value was calculated using Atwater's specific factors according to the following expression Equation (5):

$$\text{Kcal per 100 g} = P = X × (4) + CD = X × (4) + G = X × (9) + DF = X × (2) \tag{5}$$

where P denotes proteins; CD denotes available carbohydrates (excluding dietary fiber); G denotes grease; and DF denotes dietary fiber.

The analyses for each of the samples obtained for each genotype of corn were carried out in triplicate.

*2.6. Mineral Element Analysis*

The fresh samples (galls) were dried at a temperature of 60 °C for 36 h. Subsequently, they were ground until obtaining a particle size of 0.2 μm. Then, dry and pulverized samples with a weight of 0.5 g were placed in eight tetrafluoromethoxy (TFM) containers, along with 9 mL of 65% $HNO_3$ (*v/v*) and 3 mL of HCl (1:1). The first microwave digestion was carried out for 5 min with a power of 700 W at a temperature of 180 °C. The second digestion was carried out at 180 °C with a power of 500 W for 10 min, which was verified with a temperature probe in a control glass. After the digestion was complete, the containers were allowed to cool, and the resulting solution was filtered to separate possible residual particles. Finally, the supernatants were transferred to flasks and made up to 25 mL with deionized water. The resulting solutions were preserved in plastic cans that were previously washed with deionized water and stored at 4 °C until analysis via inductively coupled plasma optical emission spectrometry (ICP-OES) to determine the mineral contents (Na, Ca, Mg, P, Fe, Zn, and Mn) of the digested samples, using the ICP of Varian [33].

The results were contrasted with the values presented in the food and agriculture organization of the united nations (FAO) report on vitamin and mineral requirements. The report produced by the joint consultation of FAO/World Health Organization (WHO) experts suggests recommended daily amounts for the consumption of mineral components in the human diet for adults aged between 10 and 65 years. The recommended values are the following: 1000 mg of Ca, 260 mg of Mg, between 4.2 and 14 mg of Zn, between 9 and 27 mg of Fe, and 34 μg of Se per day [34].

## 2.7. Data Analysis

An analysis of variance (ANOVA) was performed to evaluate the productive characteristics of the net weight, leaf weight, grain weight without galls, and grams of huitlacoche produced by infected corn cobs (GMI), as well as for the analysis of minerals present in the galls of different corn genotypes. Later, Tukey's multiple mean comparison test ($p < 0.05$) was applied to determine if there were significant differences between the variables mentioned above. For the incidence rate (IP), the severity index (ISE), the chemical analysis of proximal elements, and the energy values, these values were expressed as percentages and transformed with angular arccosine $\sqrt{x} + 1$ to perform an analysis of variance with a significance of $p < 0.05$. In addition, the data were analyzed using a quadratic response surface regression model for the potential yields per hectare of huitlacoche and tests of homogeneous groups at a significance level of $p < 0.05$ using SPSS Statistics version 17 (Statistical Package for the Social Sciences) for Windows.

## 3. Results

The characteristic tumors of *U. maydis* appeared 30 days post-inoculation (Figure 1) in the 2000 plants that were inoculated. These characteristics were not observed in the control group.

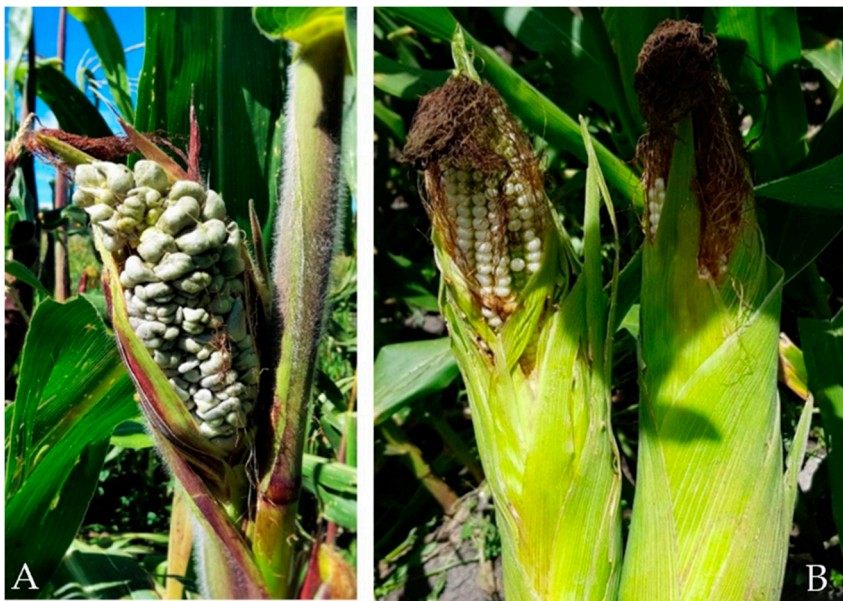

**Figure 1.** (**A**) Characteristic tumors of *U. maydis* with the white corn genotype; (**B**) control group with the Creole white genotype without the presence of tumors.

Moreover, the results present significant difference ($p < 0.05$) in terms of the production characteristics of *U. maydis* (strain MA-Um) with the different genotypes of corn evaluated here (Table 2). However, it can be seen that the Creole white genotype had the highest net weight of 515.3 g. The totomoxtle (bracts or husks) did not present significant differences for any treatment. For the weights of the corn cob grains without galls, these presented significant differences ($p < 0.05$) for the Creole white genotype with respect to the other treatments with 116.20 g; however, the Asgrow Hawk genotype presented the lowest weight of corn cob grains without galls (12.70 g), followed by the Hartz seed Z-60 genotype.

In relation to the gall weights produced by infected corn cobs (GMI), the Creole creamy-white genotype presented the best result with 249.7 g, followed by the Asgrow Hawk hybrid genotype with 220.1 g. The genotypes white, Z-60, and As-722 did not present significant differences ($p = 0.23$) between them; however, the Creole yellow, Creole blue, and Asgrow Sparrow-hawk hybrid obtained the lowest values in this investigation.

**Table 2.** Weight of corn cobs infected by the MA-Um1 strain in Nopalucan, Puebla, Mexico.

| Genotype of Corn | Characteristics | | | | |
|---|---|---|---|---|---|
| | Net Weight (g) | Sheet Weight (g) | Cob Weight (g) | Corn Kernels without Galls (g) | GMI * (g) |
| Creole white | 515.30 ± 43.9 [a] | 107.20 ± 19.6 [a] | 101.90 ± 17.3 [a,b,c] | 116.20 ± 0.9 [a] | 190.00 ± 0.4 [c] |
| Creole white-creamy | 499.00 ± 3.43 [a,b] | 80.50 ± 10.2 [a,b] | 137.00 ± 17.7 [a] | 31.80 ± 0.5 [e] | 249.70 ± 0.5 [a] |
| Creole yellow | 348.20 ± 26.1 [c,d] | 52.50 ± 8.1 [b] | 67.50 ± 11.6 [b,c] | 78.10 ± 0.7 [c] | 150.10 ± 0.4 [d] |
| Creole blue | 300.00 ± 41.6 [d] | 67.20 ± 10.1 [a,b] | 51.60 ± 6.2 [c] | 29.00 ± 0.8 [f] | 152.20 ± 0.4 [d] |
| Asgrow Hawk | 439.00 ± 27.7 [a,b,c] | 87.50 ± 6.4 [a,b] | 118.70 ± 13.4 [a,b] | 12.70 ± 0.1 [g] | 220.10 ± 0.5 [b] |
| Asgrow Sparrow-hawk | 475.60 ± 36.4 [a,b,c] | 97.90 ± 12.7 [a,b] | 130.00 ± 10.2 [a] | 91.80 ± 0.3 [b] | 155.90 ± 0.5 [d] |
| Aspros AS-722 | 479.50 ± 24.6 [a,b,c] | 100.30 ± 30.7 [a,b] | 133.80 ± 7.5 [a] | 65.80 ± 0.8 [d] | 179.60 ± 0.5 [c] |
| Hartz seed Z-60 | 364.80 ± 26.7 [b,c,d] | 61.90 ± 9.0 [a,b] | 73.20 ± 14.0 [b,c] | 29.30 ± 0.9 [f] | 200.40 ± 0.4 [b,c] |

Means followed by the same letter are not significantly different for $p \leq 0.05$ according to Tukey's test. * GMI = grams of huitlacoche produced per infected corn cob.

The most representative degree of severity obtained for the eight corn genotypes was grade 4 (Figure 2), which ranged between 43% and 74%, including 182 plants for the Asgrow Hawk genotype and 43 plants for Creole blue, respectively, obtaining significant differences ($p = 0.004$). Regarding the percentage incidence (PI), there were statistically significant differences (Figure 3) between the eight corn genotypes. The highest PI was obtained for the Hartz seed genotype Z-60 (82.25%), followed by Asgrow Hawk, Creole white, and Creole creamy-white, although not presenting significant differences between them ($p = 0.62$). However, the genotype with the lowest incidence percentage was Asgrow Sparrow-hawk (63.31%), followed by AS-722 (65.8%) and the Creole yellow and Creole blue genotypes (66.2 and 69.8%, respectively).

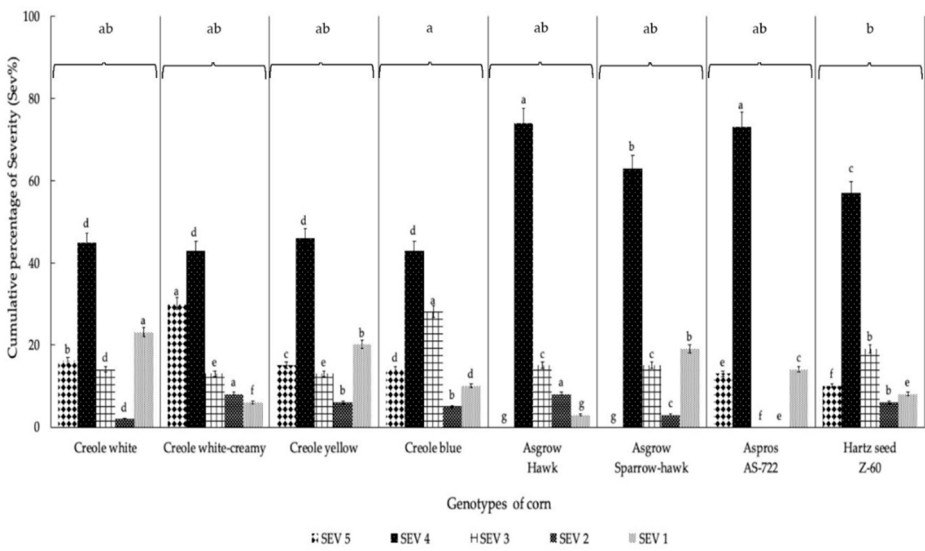

**Figure 2.** Accumulated degrees of severity (expressed as percentages) of infection of the *U. maydis* strain MA-Um1 for each genotype of corn in Nopalucan, Puebla, Mexico. Means followed by the same letter are not significantly different ($p \leq 0.05$) according to Tukey's test.

For the severity index (ISE), the values obtained for the Creole creamy-white treatment were highly significant ($p = 0.01$) compared to other treatments, obtaining a value of 64.14% (Figure 3). The Asgrow Sparrow-hawk genotype represented the lowest ISE among the genotypes with 55%.

When performing the multiple regression analysis with the experimental data for the potential yield per hectare (RPH), a second order polynomial model was obtained:

$$Y1 = 2557 + 439.0(X_1) - 61.7(X_2) - 3633(X_3) - 51.11(X_1)^2 + 1.993(X_2)^2 + 194.3(X_3)^2 + 0.205(X_1 \times X_2) + 51.1(X_1 \times X_3) + 27.1(X_2 \times X_3)$$

where Y1 is the predicted potential yield per hectare (RPH) of *U. maydis* and $X_1$, $X_2$, and $X_3$ are the parameters coded for the corn genotype, incidence percentage, and $SEV_i$, respectively. The results of the regression analysis for the response surface for the model are shown in Table 3.

**Table 3.** Response surface regression analysis of the potential yield per hectare (RPH) vs. the corn genotype, incidence percentage (PI), and severity measurement ($SEV_i$).

| Origin | DF | SS Tight | MS Tight | F-Value | *p*-Value |
|---|---|---|---|---|---|
| Model | 9 | 37,281,510,943 | 4,142,390,105 | 2441.79 | <0.001 |
| Linear | 3 | 17,898,430,762 | 5,966,143,587 | 3516.82 | <0.001 |
| $X_1$ = Genotype of corn (1, 2, … … 8). | 1 | 95,671,267 | 95,671,267 | 56.39 | <0.001 |
| $X_2$ = PI | 1 | 104,228,713 | 104,228,713 | 61.44 | <0.001 |
| $X_3$ = $SEV_i$ (1, 2, 3, 4 y 5). | 1 | 5,308,899 | 5,308,899 | 3.13 | <0.077 |
| Square | 3 | 181,098371 | 60,366,124 | 35.58 | <0.001 |
| $X_1^2$ | 1 | 79,006,159 | 79,006,159 | 46.57 | <0.001 |
| $X_2^2$ | 1 | 28,362,348 | 2,836,2348 | 16.72 | <0.001 |
| $X_3^2$ | 1 | 43,519,284 | 43,519,284 | 25.65 | <0.001 |
| Interaction of 2 factors | 3 | 51,679,520 | 17,226,507 | 10.15 | <0.001 |
| $X_1 \times X_2$ | 1 | 95,710 | 95,710 | 0.06 | 0.812 |
| $X_1 \times X_3$ | 1 | 8,809,578 | 8,809,578 | 5.19 | <0.023 |
| $X_2 \times X_3$ | 1 | 6,307,372 | 6,307,372 | 3.72 | 0.054 |
| Error | 1990 | 3,375,951,088 | 1,696,458 | | |
| Lack of fit | 1462 | 3,375,951,087 | 2,309,132 | $1.86 \times 10^{10}$. | |
| Pure error | 528 | 0 | 0 | | |
| Total | 1999 | 40,657,462,030 | | | |
| S = 1302.48 | | | | | |
| $R_2$ = 91.7% | | | | | |
| Square Fit $R_2$ = 91.66% | | | | | |
| Predictive Square $R_2$ = 91.6% | | | | | |

DF = Degrees of freedom; SS tight = tight sum of square; MS tight = tight mean square.

For the potential yield per hectare (RPH), there were highly significant differences between the treatments ($p$ = 0.001), where the Creole creamy-white genotype showed higher production (12,759.21 Kg ha$^{-1}$), presenting 75% of first quality galls. The Asgrow Hawk hybrid was characterized by generating the second highest production, with 73% of first quality galls. The lowest production was presented for the Creole yellow genotype (7167.8 Kg ha$^{-1}$), representing 20% of $SEV_1$ plants (plants without the presence of galls). We can see that the yields for the eight genotypes of corn range from 7167.8 to 12,759.21 Kg ha$^{-1}$ (Figure 4); however, if we consider the average yield between the hybrid and Creole genotypes, the overall group of hybrid genotypes presented 50.64% higher yields than the yields presented by the overall group of Creole genotypes (49.35%).

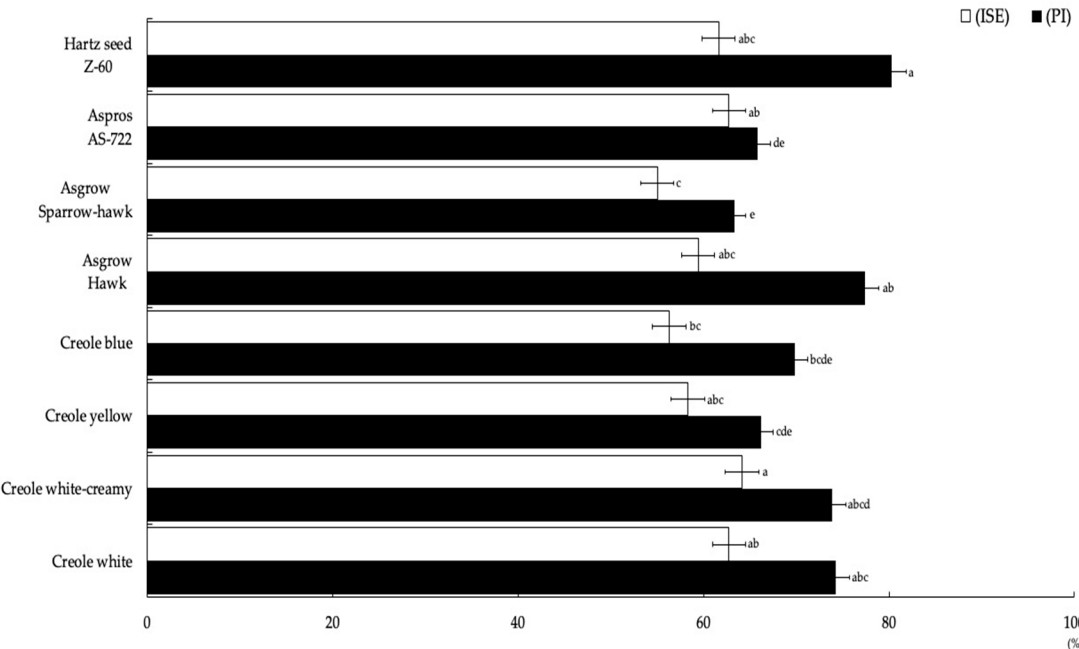

**Figure 3.** Incidence percentage and severity index values presented by *U. maydis* strain MA-Um1 for each genotype of corn in Nopalucan, Puebla, Mexico. Means followed by the same letter are not significantly different ($p \leq 0.05$) according to Tukey's test.

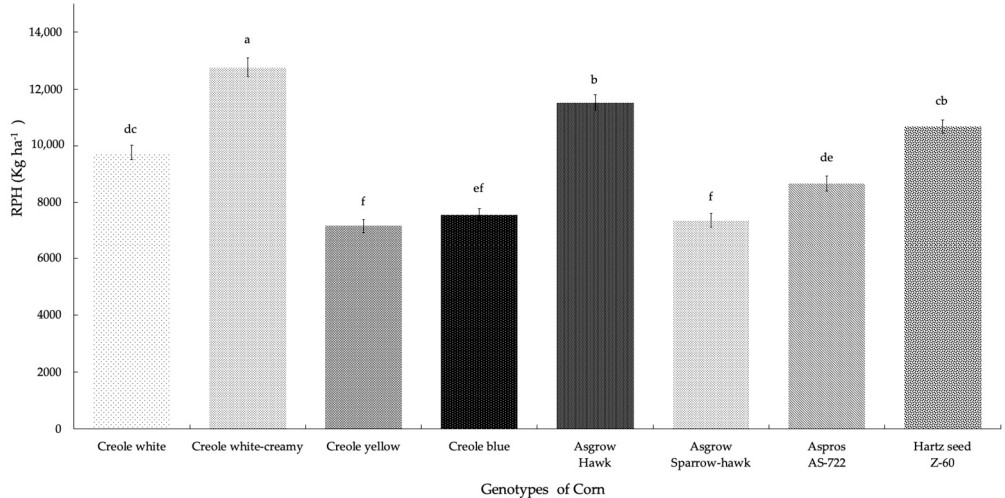

**Figure 4.** Potential yield (Kg ha$^{-1}$) of huitlacoche for each genotype of corn in Nopalucan, Puebla, Mexico. Means followed by the same letter are not significantly different ($p \leq 0.05$) according to Tukey's test.

The bromatological results for each of the corn genotypes are presented in Table 4. Higher levels of crude protein were detected in the Creole white genotype (15.3%) and the Aspros AS-722 genotype (14.6%), presenting significant differences ($p = 0.02$) with the other corn genotypes; however, the percentage for the lowest protein content fluctuated between 10.1% for the Creole blue genotype, 10.2% for the Creole creamy-white genotype, and 10.6% for the Asgrow Hawk hybrid. Regarding the ethereal extract (EE) contained in the galls of *U. maydis*, the Creole creamy-white and Creole white genotypes presented the highest contents, with 3.3 and 3.2%, respectively. It should be mentioned that the Asgrow Hawk genotype obtained the lowest EE content of 0.6%. Further significant differences ($p = 0.04$) were found in terms of the crude fiber (FC), with an average percentage of 0.14%.

**Table 4.** Bromatological analysis of the galls obtained for the different genotypes of corn in Nopalucan, Puebla, Mexico.

| Genotype of Corn | Total Humidity | Crude Protein | Ethereal Extract | Crude Fiber | Carbohydrates | Kcal per 100 g (Dry Base) | Kcal per 100 g (Fresh Base) |
|---|---|---|---|---|---|---|---|
| | (%) | (%) | (%) | (%) | (%) | (g) | (g) |
| Creole white | 90.20 ± 0.32 [a] | 15.30 ± 0.20 [a] | 3.20 ± 0.16 [a] | 0.12 ± 0.02 [g] | 76.20 ± 0.02 [h] | 395.30 ± 0.43 [a] | 38.60 ± 0.26 [b] |
| Creole white-creamy | 90.40 ± 0.50 [a] | 10.20 ± 0.28 [d] | 3.30 ± 0.23 [a] | 0.12 ± 0.01 [g] | 81.20 ± 0.05 [d] | 395.70 ± 0.76 [a] | 37.90 ± 0.37 [c] |
| Creole yellow | 89.90 ± 0.30 [a] | 12.80 ± 0.40 [c] | 2.20 ± 0.04 [c] | 0.19 ± 0.02 [e] | 79.40 ± 0.04 [e] | 389.70 ± 0.91 [b] | 39.20 ± 0.05 [a] |
| Creole blue | 89.60 ± 0.17 [a] | 10.20 ± 0.16 [e] | 0.80 ± 0.04 [f] | 0.15 ± 0.02 [f] | 83.70 ± 0.02 [a] | 383.30 ± 0.37 [e] | 39.60 ± 0.34 [a] |
| Asgrow Hawk | 89.50 ± 0.15 [a] | 10.60 ± 0.11 [e] | 0.60 ± 0.06 [g] | 0.67 ± 0.02 [b] | 83.40 ± 0.01 [b] | 382.20 ± 0.24 [f] | 39.90 ± 0.14 [a] |
| Asgrow Sparrow-hawk | 90.70 ± 0.24 [a] | 11.30 ± 0.04 [d] | 1.80 ± 0.08 [d] | 0.48 ± 0.02 [c] | 81.50 ± 0.01 [c] | 388.40 ± 0.14 [c] | 36.00 ± 0.12 [e] |
| Aspros AS-722 | 90.20 ± 0.17 [a] | 14.60 ± 0.18 [b] | 2.50 ± 0.02 [b] | 0.70 ± 0.01 [a] | 77.00 ± 0.02 [g] | 390.50 ± 0.42 [b] | 37.90 ± 0.21 [c] |
| Hartz seed Z-60 | 90.30 ± 0.011 [a] | 14.30 ± 0.22 [b] | 1.50 ± 0.02 [e] | 0.44 ± 0.02 [d] | 78.90 ± 0.02 [f] | 386.60 ± 0.51 [d] | 37.10 ± 0.18 [d] |

Means followed by the same letter are not significantly different for $p \leq 0.05$ according to Tukey's test.

The highest calorie content per 100 g of fresh material was recorded in the galls produced by the Asgrow Hawk genotype, registering a content of 39.9 Kcal per 100 g, presenting differences between the other treatments. The Asgrow Sparrow-hawk hybrid genotype presented a value of 36 Kcal per 100 g, presenting the lowest fresh content of galls resulting from the inoculation with the MA-Um1 strain. The percentages of minerals obtained for each of the genotypes are included in Supplementary Table S1, where the results show significant differences by means of Tukey's test ($p < 0.05$). The Asgrow Hawk hybrid obtained the highest concentration of Ca (25 mg per 100 g). The lowest concentration was observed in the Creole white genotype, with 19 mg per 100 g for Mg. The highest concentration was presented for the Creole yellow genotype, with 80 mg per 100 g for Mg, presenting significant differences between the other treatments; however, the Hartz seed Z-60 hybrid presented the lowest percentage of Mg in the present study. For Zn, the Aspros AS-722 hybrid obtained the highest percentage of Zn (2.3 mg per 100 g), followed by the Creole creamy-white genotype and the Asgrow Hawk hybrid, both with 2.2 mg per 100 g. The lowest concentration of Zn was presented in the Asgrow Sparrow-hawk hybrid, with 1.3 mg per 100 g, showing significant differences in Tukey's test ($p < 0.05$). In the case of Fe, the Creole creamy-white genotype presented the highest concentration with 2.8 mg per 100 g. Asgrow Hawk obtained the lowest concentration of Fe (1.4 mg per 100 g), presenting significant differences in Tukey's test ($p < 0.05$).

## 4. Discussion

The practice of including edible mushrooms in the diet of humans has prevailed due to their characteristic taste and odor. However, in recent years, interest in edible mushrooms has intensified as they constitute an important source of nutrients, proteins, vitamins, and minerals [35]. In the artificial production of huitlacoche, the symptoms that validate an adequate inoculation are mainly the formation of tumors or galls, although chlorosis, distortions, plant dwarfism, and the accumulation of anthocyanins also occur [36]. The production of huitlacoche depends on factors such as environmental conditions, the stage of development of the corn plant, the genetic susceptibility of the seed, and the pathogenicity of the strain used [5,13,19]. In the present investigation, the characteristic tumors of huitlacoche were presented 30 days after inoculation, agreeing with the work described by Ruiz-Herrera [36]. This may be due to the life cycle of *U. maydis* to include a diploid mitotic stage that corresponds to the rapid enlargement of the tumor and the conversion of the plant into fungal biomass [37].

Environmental conditions can influence the development of infection, particularly during host penetration and infection. Martínez [38] and Aydoğdu [39] reported that the most favorable temperature for development varies from 27.2 to 30.7 °C, with a relative humidity between 72 and 80%. During the development of the crops considered here, the temperature fluctuated between 16 and 32 °C with 80% relative humidity on average, and rainfall of 110 mm, and these were favorable parameters during the growth season and supported the development of *U. maydis* during the 2018 seasonal agricultural cycle [16,17].

Huitlacoche is produced commercially with some sweet genotype cultivars [5,10]; however, Aydoğdu [39] discovered that open sky conditions were favorable for the cultivation of the corn for the purpose of huitlacoche production with the *Zea mays* var. Indentata genotypes Ada-523, Pioneer-3394, and Side, as well as *Zea mays* var. indurata with the Karaçay and Karadeniz Yıldızı genotypes. These genotypes yielded more huitlacoche than the sweet corn genotypes (*Zea mays* var. saccharata), Merit and Vega genotypes, and the *Zea mays* var. everta genotype Antcin-98.

The dominant $SEV_i$ value in the inoculation process with the MA-Um1 strain for the eight genotypes evaluated here was $SEV_4$, with a fluctuation from 42% to 72% of the total plants evaluated. The hybrid corn genotypes presented greater severity in grade 4, surpassing the Creole genotypes by 25% in the present investigation. In this sense, Aguayo-González [4] reported that the ISE values generated by each strain of huitlacoche are not totally related to the genotype or color of the corn, which indicates that even when the hybrid corn genotypes are the most susceptible, depending instead on the virulence of the given strain. On the other hand, Madrigal-Rodriguez et al. [24] obtained a severity index of 90% for the H-58 hybrid with 62,500 plants per ha, showing results higher than those obtained in the present study. However, in the same investigation, the Cóndor genotype presented 59.38% severity, showing lower results than those obtained in the present investigation. Here, the total average value for the ISE was 69%, with significant differences between the eight genotypes evaluated.

Regarding the PI, it was observed that the MA-Um1 strain of *U. maydis* has high infective potential, given that all the genotypes studied presented some degree of infection that resulted in a value for the PI between 79 and 88.70%. Salazar-Torres et al. [40] obtained higher PIs with the hybrids Cobra (85.8%), Oso (84.5%), and A7573 (78.5%) and the Creole R12 (85%), B7 (83.3%), and B3 (73%) genotypes, where the rest of the Creoles presented PI values less than 70%. Likewise, Valdez-Morales et al. [15] evaluated the production of huitlacoche for 15 Creole genotypes and reported PI values that ranged from 30.9% to 92%, with data that are equivalent to those obtained for the open field evaluation carried out in the present investigation. In the present investigation, for the PI, significant differences were found, however the Hartz seed Z-60 genotype presented the highest PI (80.25%).

Valdez-Morales et al. [15] mentioned that they obtained yields of up to 15 tons ha$^{-1}$ with a Creole corn genotype via using an artificial inoculation method in controlled conditions. In the present investigation, the Creole creamy-white genotype showed higher production (12.75 tons ha$^{-1}$), presenting 75% of first-quality galls; however, the eight genotypes of corn presented optimal yields with potential productive value, with results similar to those reported by Aguayo-González et al. [4]. Likewise, Martínez et al. [16] obtained yields of 9.1, 8.4, 8.2, and 8 tons ha$^{-1}$, respectively, from 300 families that were evaluated based on a sample of 19 plants inoculated at a density of 60,000 plants per hectare. Pataky and Chandler [9] obtained a yield of 131 g of huitlacoche per cob of corn, inoculated six days after the appearance of the silk channel, with results that are inferior to those obtained in the present investigation.

Mushrooms contain 90% water, 1% to 4% protein, 0.2% to 0.8% fat, 0.3% to 2.8% carbohydrates, 0.3% to 7.0% fiber, and 0.6% to 1% ash, along with potassium, calcium, phosphorus, magnesium, iron, zinc, and copper as the most common minerals [18,34]. The percentage of protein per 100 g of dry matter ranged between 15.3% for Creole white and 10.2% for Creole blue, presenting significant differences. These results are similar to those found by Valdez-Morales et al. [15], where they mentioned the protein percentage in huitlacoche to be 11.3% on average. In the same way, Aydogdu and Golukcu [32] mentioned that huitlacoche has a considerable proportion of protein (approximately 12% on a dry basis). Pimentel et al. [8] obtained 13.4% protein for the Bengal corn genotype, showing higher results than those obtained in the present investigation; however, for the QPM corn genotype (High-Quality Protein Maize), 11.1% protein was presented, showing similar results to the Asgrow Sparrow-hawk genotype in the present investigation. Valdez-Morales et al. [15] found a considerable amount of crude protein in Creole corn (9.8%) and 11.3% in hybrid corn.

An average ethereal extract of 2% (dry base) was found in the present study, with results similar to those reported by Beas [41], where they presented samples from eight genotypes of corn collected

in different localities in Aguascalientes and Jalisco, reporting ether extracts between 2.4% and 3.6%. Likewise, Paredes et al. [3] reported the fat content for huitlacoche to be between 2.7% and 6.5%, presenting higher results than those reported in this research.

Aydogdu and Golukcu [32] mentioned that huitlacoche has a considerable amount of carbohydrates (45% on a dry basis); however, in the present study, the resulting carbohydrate contents for the eight genotypes present an average value of 80.2% on a dry basis. The data here correspond to the data mentioned by Venegas et al. [14], who reported a value that fluctuates between 77% to 82.7% for 14 hybrid genotypes of sweet corn.

The highest calorie content was recorded in the galls produced by the Asgrow Hawk hybrid, presenting 39.9 Kcal per 100 g of fresh galls, followed by the Creole blue genotype. The Asgrow Sparrow-hawk genotype obtained the lowest content of Kcal per 100 g on a dry basis for the galls produced by the MA-Um1 strain.

The determinations of minerals present in the eight genotypes here show variation between the hybrid and Creole genotypes; however, the Asgrow Hawk hybrid obtained the highest concentration of Ca (25 mg per 100 g). Aydogdu and Golukcu [33] mentioned the contents of calcium and magnesium to be 18.6 and 26.26 mg per kg, respectively, which is higher than the results obtained in the present investigation. In addition, the highest concentration of magnesium was presented by the Creole yellow genotype with 80 mg per 100 g, and the Zn content was kept in 1.8 mg per 100 g, which is close to that reported by Turlo et al. [42], who mentioned that a Zn content of approximately 5 mg supports the growth and development of *Lentinus edodes*. *U. maydis* makes a large number of nutrients available, including metals such as zinc, which is notable when considering that Zn deficiency is a serious problem throughout the world [43].

*Ustilago maydis*, like other edible mushrooms (*Agaricus*, *Boletus*, *Ramaria*, *Lactarius*, and *Pleurotus*), has high water (91%), protein (15%), and mineral (10%) contents. Additionally, the Zn and Fe contents are notable, with averages of 2 and 2.1 mg per 100 g, respectively, which are considerable values in terms of the daily intake recommended by the FAO (Table S1) [34].

## 5. Conclusions

The Creole creamy-white corn genotype showed the highest production yield here (12,759.21 Kg ha$^{-1}$), representing 43% of galls with a SEV$_i$ value of 4 and with 30% with a SEV$_i$ value of 5, followed by the Asgrow Hawk hybrid with 11,670.3 Kg ha$^{-1}$, representing 74% of galls with a SEV$_i$ value of 4, which was explained by the higher incidence rates (PI) of 73.90% and 77.38% and the severity index (ISE) values of 64.14% and 59.41%, respectively.

The highest calorie content was registered in the galls produced by the Asgrow Hawk genotype with a value of 39.9 Kcal per 100 g of fresh mushroom. The Asgrow Hawk hybrid obtained the highest concentration of Ca (25%). The Aspros AS-722 hybrid obtained the highest percentage of Zn (2.3%), and the Creole creamy-white genotype presented the highest concentration of Fe (2.8%).

The results obtained with the MA-Um1 strain are promising and open possibilities for cultivating this edible mushroom with two high-potential genotypes for cultivation in seasonal conditions in open fields. The Creole creamy-white and Asgrow Hawk genotypes have the potential to drive technological development in terms of huitlacoche production in Nopalucan, Puebla, Mexico.

**Supplementary Materials:** The following are available online at http://www.mdpi.com/2077-0472/10/11/535/s1, Table S1. Mineral content present for each genotype evaluated.

**Author Contributions:** Conceptualization, O.G.R., I.T.S. and O.R.A.; methodology, I.T.S., O.G.R. and G.M.; software, M.d.l.A.V.d.I. and O.R.A.; validation, G.M., I.T.S. and O.R.A.; formal analysis, C.P.L., I.T.S. and O.R.A.; resources, I.T.S. and O.R.A.; Original-draft preparation, O.G.R. and O.R.A.; writing—review and editing, O.G.R. and O.R.A.; visualization, O.R.A., I.T.S.; supervision, G.M.; project administration, O.R.A.; funding acquisition, O.R.A. All authors have read and agreed to the published version of the manuscript.

**Funding:** This research was supported by the program BECAS-TESIS-CONACyT-project 578 and the Secretary of Public Education (SEP) in the PRODEP 2020 program.

**Acknowledgments:** The authors are grateful to National Council of Science and Technology (CONACYT) and Postgraduate Master Program in Sustainable Management of Agroecosystems at Benémerita Universidad Autónoma of Puebla. Likewise, to the Ejido of Nopaluca and its president Rafael Lopez Gallegos, for the establishment of the experiment in their agricultural plots and the collaboration of producers involved in this research.

**Conflicts of Interest:** The authors declare no conflict of interest.

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
