# Peer review of "Evaluation of Eight Genotypes of Corn for the Commercial Cultivation of Huitlacoche in Nopalucan, Puebla, Mexico"

_agriculture, doi:10.3390/agriculture10110535_

Round 1

Reviewer 1 Report

The authors Rahme et al., describe in this manuscript and interesting evaluation and characterization of different genotypes of corn used for commercial cultivation of Huitlacoche, considered an exotic dish of Mexican cuisine.

During 2018, after an artificial inoculation with the Ustilago maydis MA-Um-1 strain, authors focused on their research on evaluating the potential galls yield per hectares, the incidence percentage, severity index and a chemical analysis and mineral  elements to the galls.

2            The author in all this manuscript describe the evaluation of eight varieties of corn. In the description of different genotypes though authors choose 4 commercial hybrids and 4 creole varieties so the evaluation is NOT of eight varieties but eight genotypes or 4 maize varieties

27          better open field

28          Some information about the average yield of varieties and hybrids in your country under particular cultivation condition at 2300  masl as in my country the hybrids production is higher

52          more information about the virulence and pathogenicity of the strain of Ustilago maydis used in artificial inoculation

61          enrich other foods: as you inoculate the silk channel of the ear with a syringe, other fungal pathogen could use this way to entry and attack the ear.

In your environmental condition there is mycotoxigenic strain of fungi that could be present in a natural field that could compromise the quality and safety of the material? Have you consider an evaluation of presence of mycotoxins produce by other fungi? Some of them could be toxigenic for feed and also for human consumption

133        in the control group of genotypes not artificially inoculated which is the percentage of presence of galls (under natural infection). Normally in your conditions which is the percentage of attack of Ustilago maydis?

If commercial hybrid are also used for kernel production, the presence of an high severity index do not influence the quality of the kernel and  flour

How is the normal production of creole white cream? It is strange observe a production higher than hybrids

Figure 3 not completed….lack one genotypes

Haven’t you consider the possibility to analyse different data in PCA to find possible correlation?

Author Response

(Reviewer 1)

The authors Rahme et al., describe in this manuscript and interesting evaluation and characterization of different genotypes of corn used for commercial cultivation of Huitlacoche, considered an exotic dish of Mexican cuisine.

Thank you for the comment about our work, as you say, it is an interesting work; since at present there are few works related to the production of Huitlacoche in Mexico under rainfed conditions, although its consumption in Mexican cuisine has existed for a long time, as we refer in the article. In addition, our work offers and validates a simple technique to be able to venture into the commercial production of Huitlacoche and diversify the offer of products of the farmers to improve their economic income from their crops in Puebla, Mexico.

During 2018, after an artificial inoculation with the Ustilago maydis MA-Um-1 strain, authors focused on their research on evaluating the potential galls yield per hectares, the incidence percentage, severity index and a chemical analysis and mineral elements to the galls.

It is right.

  1. The author in all this manuscript describe the evaluation of eight varieties of corn. In the description of different genotypes though authors choose 4 commercial hybrids and 4 creole varieties, so the evaluation is NOT of eight varieties but eight genotypes or 4 maize varieties.

Its observation is considered valid, and it is incorporated into the work, marked in blue.

  1. Better open field.

Its observation is considered valid, and it is incorporated into the work, marked in blue.

  1. Some information about the average yield of varieties and hybrids in your country under particular cultivation condition at 2,300 masl as in my country the hybrids production is higher.

In this section we are referring to the potential production of Huitlacoche that can reach in one hectare in different maize genotypes and in open field conditions. We are not referring to corn production; However, in line 69-70 we mention the corn production data for Nopaluca from the farm, which is 17,560 tons in 7,689 hectares in open field conditions, with an average yield of 2.28 tons ha-1. This yield is low, but in our country the average yield for corn (grain) is 2.2 tons ha-1 (SIAP 2020), but under irrigation conditions it increases to 7.5 tons ha-1, however, 60% of the Agriculture in Mexico is in open field conditions. That is why this work opens the opportunity to produce Huitlacoche under these conditions and obtain more economic resources, since unlike corn, Huitlacoche has a better price in the national market.

  1. More information about the virulence and pathogenicity of the strain of Ustilago maydis used in artificial inoculation.

The MA-Um1 strain of U. Maydis is the result of a previous work (under review for publication) to obtain diploid yeasts presumptive of basidiospores from the combination of 2 native varieties from the state of Morelos, Mexico and 6 from the state of Puebla. Where the MA-Um1 strain from the P4-M1 cross presented the highest percentage of infection (78%) in corn grains under in vitro conditions. That is why it was selected for field tests and in the state of Puebla.

The text is modified and data on the strain MA-Um1 strain of U. Maydis is added.

It is worth mentioning that the other reviewer presented the same observation, and it was made in green colour.

  1. Enrich other foods: as you inoculate the silk channel of the ear with a syringe, other fungal pathogen could use this way to entry and attack the ear.

The text is modified, and a more current appointment is added.

Huitlacoche has been characterized as a high quality nutraceutical fungus, as it contains β-glucans, free sugars, antimutagenic substances [15], in addition to containing significant amounts of ferulic acid, quercetin, ergosterol, linoleic and oleic acids, as well as large amounts of magnesium, phosphorus, calcium, sodium and high protein content; characteristics with excellent antioxidant potential and important health benefits [16], where it is consumed mainly in Mexico, Central America and the United States, where the cost of huitlacoche for the restaurant market can range between 30 and 40 (USD) per kilogram of casing fresh [11, 18,19].

In your environmental condition there is mycotoxigenic strain of fungi that could be present in a natural field that could compromise the quality and safety of the material? Have you consider an evaluation of presence of mycotoxins produce by other fungi? Some of them could be toxigenic for feed and also for human consumption

Interesting observation, however, in this study the objective of the work was different.

But we can mention that in natural communities there are very diverse interactions between microbial symbionts that can be characterized as parasites or as mutualists, in the case of U. maydis, it is an endophytic fungus and it is not so easy to observe and qualify these interactions with other organisms; because they do not cause visible symptoms in their presence within their host. There is a study with Fusarium verticillioides and U. maydis, where they evaluate the fungus-fungus interactions in the growth of the endophyte and pathogen within the plant and see their effect on the growth of the plant. Where they conclude that:

F. verticillioides modulates the growth of U. maydis and therefore reduces the aggressiveness of the pathogen towards the plant, they also mention that F. verticillioides can decompose the plant compounds that limit the growth of U. maydis and obtain a benefit, therefore, an endophyte such as F. verticillioides can function as a defensive mutualist and a parasite, expressing nutritional modes that depend on the ecological context, which is another factor to consider.

But we can mention that young galls are the ones that have the best characteristics to be consumed without any risk to health, as is the case in this study and many others that are studying their production.

  1. In the control group of genotypes not artificially inoculated which is the percentage of presence of galls (under natural infection). Normally in your conditions which is the percentage of attack of Ustilago maydis?

The control group for this experiment indicated that there was no presence of U. maydis in the study region in the eight genotypes studied. However, there are no data on the genetic materials of susceptible maize or on the incidence of infection by U. maydis by region. That is why this study was carried out for the region of Nopaluca-Puebla, Mexico.

But we can mention that in a recent stage (2016) the natural production of Huitlacoche was evaluated in the state of Aguascalientes, Mexico, obtaining the highest natural production in a genotype of white creole corn with 90,554 kg ha-1 with 18% severity. Results lower than those reported for artificial inoculation.

However, in line 42, the production of huitlacoche in a natural way is mentioned, where it is mentioned that it can reach up to 500 tons annually during July and August in the markets of Mexico City. That is why our research is carried out in times of secano or open field conditions.

If commercial hybrid are also used for kernel production, the presence of an high severity index do not influence the quality of the kernel and flour.

Interesting question, however, it was not considered for this work. But I can clarify your doubt, I think you are thinking that the corn that was inoculated both in hybrids and in creoles where the galls developed different levels of severity, may affect the quality of the grain that was not affected by U. maydis. At this point it should be clarified that in those corn grains that still remained on the cob they were smaller, however, in Mexican food and when preparing a dish with Huitlacoche, they are also used in the preparation of food. But we can take your proposal for another investigation.

That is why we tried to see if there were differences in the amount of protein and minerals in the galls produced in different maize genotypes, and as can be seen in table 4, the hybrid genotypes present a greater amount of protein to exception of the genotype white and yellow creole but the difference is tiny.

How is the normal production of creole white cream? It is strange observe a production higher than hybrids.

I agree with you, but we have to clarify that we are talking about Huitlacoche production, not corn grain, they are two different issues. Some authors mention that there is better production of Huitlacoche in creole genotypes, since some hybrid genotypes may present greater resistance to the attack of U. maydis. In the present investigation the differences were minimal, that is why it is important to carry out these research works, to know the potential of various genotypes for the commercial production of Huitlacoche in the open field.

Figure 3 not completed….lack one genotypes

This is figure 3. And there are the 8 genotypes, (page 8). Check.

Haven’t you consider the possibility to analyse different data in PCA to find possible correlation?.

In the present investigation we opted to validate the data by the response surface regression analysis for the potential yield (RPH) vs. the variables: corn genotype, PI and SEVi. that are mentioned in the job objectives.

The analysis gave us a high significant difference in the variable maize genotype and PI, however, in the variable SEVi. no differences were found. However, when we see the interaction of two variables; we observed that the maize genotype interaction and SEVi. they present highly significant differences (0.023) with the potential yield, which was the main objective of the research.

We believe that the response surface regression analysis for this work conforms to our experimental design, however we can consider your proposal for another investigation, since if we perform the principal component analysis, the results could vary and that leads us to modify The text of document and the editorial team gave us only 8 days to make all the comments from the reviewers.

___________________________________________________________________

I hope have answered all your questions and comments about the proposed work, I also appreciate the time invested in the review and improvement of this research document.

___________________________________________________________________

Reviewer 2 Report

In their manuscript, the authors compared different varieties of corn for their suitability of production of Huitlacoche by infection of cobs with a culture of Ustilago maydis in a field experiment conducted in Nopalucan de la Granja, Puebla, Mexico.

Although I understand the difficulties of summarizing research results in a foreign language, and I acknowledge the efforts of the authors to present their results in English, the current level of English language and style is not sufficient for publication. I have the impression, that a text that was composed originally in Spanish has been translated by an automatic translation program and was not or only insufficiently proofread after this first translation draft. I therefore recommend to revise the complete manuscript by a native English speaker that has knowledge either of the Spanish language or of the research that was conducted. 

I will give below a few (non-exhaustive) examples of where I see problems in the text.

It seems to me that “Nopalucan de la Granja” is a name of a place, which means that the middle words “de la” should not be translated into English.

Some Spanish words are still in the text (e.g. secano), the words gills, galls and guts seem all to be used to mean the same thing, at some positions the area is given in per hours instead of per hectare, there are several translation problems (e.g. line 45 channel silk, leaves shell, emergence silks), which make it impossible to understand the sentence, some words appear to be out of context (e.g. media sedimentation (line 47) or selective production (line 48)).

It seems to me that the strain MA-Um1 that was used in this experiment is in fact not a strain but a mixture of mating compatible strains. It is well-known that haploid strains of U. maydis are incapable of plant infection and also do not produce filaments on plates. This should be specified, or if it is indeed a single but diploid strain than this should be mentioned.

The sentence line 92/93 is not understandable. Which transition was made? Between what? Of what? What has the pH to do with this and why was chloramphenicol added?

I understand that to generate the inoculum, two flasks were inoculated whereby one flask was used to determine the concentration of basidiospores, whereas the other was ASSUMED to have the same concentration of basidiospores and was used as inoculum. This is not correct doing. If you want to know the concentration of basidiospores used as inoculum, of course the material used as inoculum should have been evaluated.

The explanation of PI% and ISE% is not precise. What means No. corn? Does it mean number of maize plants for PI% and number of cobs for ISE%? Define total infected corn: plants that have at least one cob with galls, or cobs with at least one gall, or total of inoculated plants? Check the number of brackets in the PI% formula, as is there are brackets missing and too much.

The sentence 203-209 needs to be completely revised.

When explaining the results, please make an effort to also state what was done to come to the results. Do not repeat the materials and methods part but describe the overall experiment so that the results part can be understood without having to work through the methods section.

The second sentence of the results claims that infection is dependent on the concentration of cells. How can that be, if only one concentration was used. If different concentrations were used, where is the data that lead to this statement? The second half of that sentence insinuates that the infection rate is influenced by environmental conditions, referencing Figure 1. Figure 1, however only seems to represent the measured minimal, maximal and medium temperatures as well as mm rainfall (in contrast to what is explained in the figure legend) and there is no correlation with infection rate that would back up the statement made. In Fig. 1 it is not clear why the bars for Aug and Sept have a different pattern or what that pattern means or if there was any rainfall in these months.

For the data evaluated in Table 2, more explanation is needed. Do I understand correctly, that the ear was separated from the plant and weighed (net weight), and that this ear was then split in three fractions, the husks (sheet weight), the galls (GMI) and the remaining cob (cob weight)? If that is so this should be explained. And if that is so, why do the three valued not add up to the net weight? Is there a fourth fraction that was not mentioned?

What exactly was measured in the last two columns?

The error bar for Figure 2, which data does it evaluate and is this shown? If it refers just to the last category, I wonder where the analysis of the other categories would be indicated. If it refers to an overall calculated value, that this value should also be given. In any case, the type of error bar should be indicated in each figure legend.

I have difficulties in believing the statistical analysis of the data presented in Figure 4: The values for creole blue and creole yellow are statistically significantly different, whereas those of creole white creamy (the highest one) and those of creole yellow (the lowest one) are not?

Check formatting of Table 4

In the discussion, I sincerely lack any statement that would answer the research question (i.e. which of the investigated varieties is best for huitlacoche production) and would indicate in how far the obtained results help in giving this answer (i.g. is weight or size or content the most important criterium).

Author Response

(Reviewer 2)

The observations modified by the reviewer 1 are marked in blue in the document. Those in green, are the observations and changes in the document suggested by you.

In their manuscript, the authors compared different varieties of corn for their suitability of production of Huitlacoche by infection of cobs with a culture of Ustilago maydis in a field experiment conducted in Nopalucan de la Granja, Puebla, Mexico.

Although I understand the difficulties of summarizing research results in a foreign language, and I acknowledge the efforts of the authors to present their results in English, the current level of English language and style is not sufficient for publication. I have the impression, that a text that was composed originally in Spanish has been translated by an automatic translation program and was not or only insufficiently proofread after this first translation draft. I therefore recommend to revise the complete manuscript by a native English speaker that has knowledge either of the Spanish language or of the research that was conducted.

Thanks for the comments.

The document was reviewed by a native speaker of the English language (USA) however it was reviewed again in the recommended scores, marked in green.

If in this answer, you believe that it is not enough, I comment that we can opt for the correction of style offered by the magazine; this to comply with the language and not be an inconvenience for the publication of these results. We believe that this document has technical-scientific relevance for the commercial cultivation of Huitlacoche in the state of Puebla, Mexico.

It seems to me that “Nopalucan de la Granja” is a name of a place, which means that the middle words “de la” should not be translated into English.

Your appreciation is correct. In this case the information was verified, and we could see an error, "Nopalucan de la Granja" is the name of the municipal seat of the municipality, the municipality is only called Nopalucan. It was corrected throughout the document marked in green.

Some Spanish words are still in the text (e.g. secano), the words gills, galls and guts seem all to be used to mean the same thing, at some positions the area is given in per hours instead of per hectare, there are several translation problems (e.g. line 45 channel silk, leaves shell, emergence silks), which make it impossible to understand the sentence, some words appear to be out of context (e.g. media sedimentation (line 47) or selective production (line 48)).

As I explained previously, the document was verified and those words in Spanish were eliminated, in addition the term galls was homogenized for the entire document and the wording of the paragraphs marked by you was improved.

It seems to me that the strain MA-Um1 that was used in this experiment is in fact not a strain but a mixture of mating compatible strains. It is well-known that haploid strains of U. maydis are incapable of plant infection and also do not produce filaments on plates. This should be specified, or if it is indeed a single but diploid strain than this should be mentioned.

The sentence line 92/93 is not understandable. Which transition was made? Between what? Of what? What has the pH to do with this and why was chloramphenicol added?

This section is better written, and it is explained that the strain is diploid as you suggested.

I also comment the following:

The MA-Um1 strain of U. Maydis is the result of a previous work (in review for publication) to obtain presumptive diploid yeasts of basidiospores from the combination of 2 native varieties from the state of Morelos, Mexico and 6 from the state of Puebla. Where the MA-Um1 strain comes from the P4-M1 cross, which presented the highest percentage of infection (78%) in corn grains under in vitro conditions. That is why this strain was selected for field tests and in the state of Puebla.

The use of chloramphenicol is to avoid contamination in the culture medium. This information is included in the section.

I understand that to generate the inoculum, two flasks were inoculated whereby one flask was used to determine the concentration of basidiospores, whereas the other was ASSUMED to have the same concentration of basidiospores and was used as inoculum. This is not correct doing. If you want to know the concentration of basidiospores used as inoculum, of course the material used as inoculum should have been evaluated.

It is correct, two flasks were used, where the first served to make adjustments to reach the concentrations in various tests and check its viability, the second was used as a mother matrix, however if viability tests were performed and the concentration was measured that was used in the field experiment. This was not put in the first document; however, the wording is improved, and this information is incorporated.

The explanation of PI% and ISE% is not precise. What means No. corn? Does it mean number of maize plants for PI% and number of cobs for ISE%? Define total infected corn: plants that have at least one cob with galls, or cobs with at least one gall, or total of inoculated plants? Check the number of brackets in the PI% formula, as is there are brackets missing and too much.

This section is better written and each of the formulas used is explained in more detail, also the parentheses of each formula are reviewed, and the errors found are corrected.

The sentence 203-209 needs to be completely revised.

The paragraph is restructured, as you suggested.

The second sentence of the results claims that infection is dependent on the concentration of cells. How can that be, if only one concentration was used. If different concentrations were used, where is the data that lead to this statement? The second half of that sentence insinuates that the infection rate is influenced by environmental conditions, referencing Figure 1. Figure 1, however only seems to represent the measured minimal, maximal and medium temperatures as well as mm rainfall (in contrast to what is explained in the figure legend) and there is no correlation with infection rate that would back up the statement made. In Fig. 1 it is not clear why the bars for Aug and Sept have a different pattern or what that pattern means or if there was any rainfall in these months.

The first paragraph is structured and this information that is not relevant for the research work is eliminated.

For the data evaluated in Table 2, more explanation is needed. Do I understand correctly, that the ear was separated from the plant and weighed (net weight), and that this ear was then split in three fractions, the husks (sheet weight), the galls (GMI) and the remaining cob (cob weight)? If that is so this should be explained. And if that is so, why do the three valued not add up to the net weight? Is there a fourth fraction that was not mentioned?

Your appreciation is correct, there is a fourth fraction that if it is measured and represents the grains of corn without galls, this information is present in severity degree 1 (figure 2) that mentions us without galls present; which is the result of small corn kernels.

However, Table 2 is complemented with this information.

What exactly was measured in the last two columns?

The last two columns of the table 2, present the data of the total length and width of the corn cob, however this information is decided to eliminate it, since it leaves the evaluation context (weight).

The error bar for Figure 2, which data does it evaluate and is this shown? If it refers just to the last category, I wonder where the analysis of the other categories would be indicated. If it refers to an overall calculated value, that this value should also be given. In any case, the type of error bar should be indicated in each figure legend.

The error bar in Figure 2 refers only to a global calculated value of the severity scale. However, it is decided to change the representation of the results of figure 2 and show the error bar by category and the global error bar.

I have difficulties in believing the statistical analysis of the data presented in Figure 4: The values for creole blue and creole yellow are statistically significantly different, whereas those of creole white creamy (the highest one) and those of creole yellow (the lowest one) are not?

Your assessment is correct, and I apologize for the error that was made in the interpretation of the data. In the written document, the interpretation of the results is correct, in this case the error in figure 4 is corrected.

Check formatting of Table 4.

The format was revised, and the errors found were corrected.

In the discussion, I sincerely lack any statement that would answer the research question (i.e. which of the investigated varieties is best for huitlacoche production) and would indicate in how far the obtained results help in giving this answer (i.g. is weight or size or content the most important criterium).

The wording of the discussion section is improved, and the expected results are discussed adequately, thus supporting the objective of this research.

___________________________________________________________________

I hope have answered all your questions and comments about the proposed work, I also appreciate the time invested in the review and improvement of this research document.

___________________________________________________________________

Round 2

Reviewer 2 Report

I only quickly scanned the manuscript. Overall, the manuscript seems to have significantly improved. However, I still think there is a mistake in the newly introduced methods where they claim at two positions that they cultivated U. maydis yeasts at 37°C, which is too high for this organism. I also advise to have the complete manuscript checked by the style check offered by the magazine as suggested by the authors.

Author Response

Second round

I only quickly scanned the manuscript. Overall, the manuscript seems to have significantly improved. However, I still think there is a mistake in the newly introduced methods where they claim at two positions that they cultivated U. maydis yeasts at 37°C, which is too high for this organism. I also advise to have the complete manuscript checked by the style check offered by the magazine as suggested by the authors.

Thanks for the feedback.

The document was sent and reviewed by the MDPI journal service. The text has been checked for correct use of grammar and common technical terms, and edited to a level suitable for his publication. Certified annex of the service.

Regarding the comment on the methodology, the cited technique was reviewed and the error regarding the temperature for its growth was confirmed. The error presented in green color is corrected.

Please do not hesitate to contact me if more information is required on this submission. I also appreciate the time invested in the review and improvement of this research document.

Thank you
